# Significant Impact of Coffee Consumption on MR-Based Measures of Cardiac Function in a Population-Based Cohort Study without Manifest Cardiovascular Disease

**DOI:** 10.3390/nu13041275

**Published:** 2021-04-13

**Authors:** Ebba Beller, Roberto Lorbeer, Daniel Keeser, Franziska Galiè, Felix G. Meinel, Sergio Grosu, Fabian Bamberg, Corinna Storz, Christopher L. Schlett, Annette Peters, Alexandra Schneider, Jakob Linseisen, Christa Meisinger, Wolfgang Rathmann, Birgit Ertl-Wagner, Sophia Stoecklein

**Affiliations:** 1Institute of Diagnostic and Interventional Radiology, Pediatric Radiology and Neuroradiology, University Medical Center Rostock, 18057 Rostock, Germany; felix.meinel@med.uni-rostock.de; 2Department of Radiology, University Hospital, LMU Munich, 81377 Munich, Germany; roberto.lorbeer@med.uni-muenchen.de (R.L.); daniel.keeser@med.uni-muenchen.de (D.K.); franziskamgalie@gmail.com (F.G.); Sergio.Grosu@med.uni-muenchen.de (S.G.); sophia.stoecklein@med.uni-muenchen.de (S.S.); 3Department of Psychiatry and Psychotherapy, Ludwig-Maximilians University Hospital LMU, 80336 Munich, Germany; 4Munich Center for Neurosciences (MCN)–Brain & Mind, 82152 Planegg-Martinsried, Germany; 5Department of Diagnostic and Interventional Radiology, Medical Center–University of Freiburg, Faculty of Medicine, University of Freiburg, 79106 Freiburg, Germany; fabian.bamberg@uniklinik-freiburg.de (F.B.); christopher.schlett@uniklinik-freiburg.de (C.L.S.); 6University Heart Center Freiburg-Bad Krozingen, 79189 Bad Krozingen, Germany; 7Department of Neuroradiology, Medical Center–University of Freiburg, Faculty of Medicine, University of Freiburg, 79098 Freiburg, Germany; corinna.storz@uniklinik-freiburg.de; 8Institute of Epidemiology, Helmholtz Zentrum München-German Research Center for Environmental Health (GmbH), 85764 Neuherberg, Germany; peters@helmholtz-muenchen.de (A.P.); alexandra.schneider@helmholtz-muenchen.de (A.S.); 9LMU Munich, IBE-Chair of Epidemiology, 85764 Neuherberg, Germany; 10German Centre for Cardiovascular Research (DZHK), Partner Site Munich Heart Alliance, 80802 Munich, Germany; 11Independent Research Group Clinical Epidemiology, Helmholtz Zentrum München, German Research Center for Environmental Health, 85764 Neuherberg, Germany; j.linseisen@unika-t.de; 12Ludwig-Maximilians Universität München, UNIKA-T Augsburg, 86156 Augsburg, Germany; christa.meisinger@helmholtz-muenchen.de; 13German Diabetes Center, Institute of Biometrics and Epidemiology, Leibniz Institute at Heinrich Heine University Düsseldorf, 40225 Düsseldorf, Germany; wolfgang.rathmann@ddz.de; 14Department of Medical Imaging, The Hospital for Sick Children, University of Toronto, Toronto, ON M5G 1X8, Canada; BirgitBetina.Ertl-Wagner@sickkids.ca

**Keywords:** coffee, magnetic resonance imaging, cardiac function, visceral adipose tissue

## Abstract

Subclinical effects of coffee consumption (CC) with regard to metabolic, cardiac, and neurological complications were evaluated using a whole-body magnetic resonance imaging (MRI) protocol. A blended approach was used to estimate habitual CC in a population-based study cohort without a history of cardiovascular disease. Associations of CC with MRI markers of gray matter volume, white matter hyperintensities, cerebral microhemorrhages, total and visceral adipose tissue (VAT), hepatic proton density fat fraction, early/late diastolic filling rate, end-diastolic/-systolic and stroke volume, ejection fraction, peak ejection rate, and myocardial mass were evaluated by linear regression. In our analysis with 132 women and 168 men, CC was positively associated with MR-based cardiac function parameters including late diastolic filling rate, stroke volume (*p* < 0.01 each), and ejection fraction (*p* < 0.05) when adjusting for age, sex, smoking, hypertension, diabetes, Low-density lipoprotein (LDL), triglycerides, cholesterol, and alcohol consumption. CC was inversely associated with VAT independent of demographic variables and cardiovascular risk factors (*p* < 0.05), but this association did not remain significant after additional adjustment for alcohol consumption. CC was not significantly associated with potential neurodegeneration. We found a significant positive and independent association between CC and MRI-based systolic and diastolic cardiac function. CC was also inversely associated with VAT but not independent of alcohol consumption.

## 1. Introduction

Coffee is one of the most widely consumed beverages worldwide [1,2,3]. Therefore, even small health effects of drinking coffee could have a major impact on public health [4,5]. Thus far, there have been mixed results whether coffee consumption is associated with general health benefits or poses a potential risk to health. Coffee consumption has been, for example, associated with a decreased risk of neurodegenerative diseases such as Alzheimer’s disease and Parkinson’s disease [6] but also depression [7], obesity [8], and type II diabetes [9,10]. However, there are conflicting results on whether intake of coffee is associated with adverse effects on blood pressure [11,12,13]^.^ Nevertheless, most studies suggest that coffee consumption is associated with adverse effects on blood cholesterol [14,15] and homocysteine levels [16]. Therefore, coffee consumption might be a risk factor for coronary heart disease. Yet, epidemiological studies, in general, suggest otherwise, even reporting a reduced risk for total mortality in general [17], and cardiovascular mortality in particular [18,19,20,21,22], associated with coffee consumption. Nevertheless, the potential pathophysiological or beneficial role of coffee intake in consumers without manifest vascular disease is not yet sufficiently understood.

Thus, the aim of the present study was to determine the relationship between coffee consumption and possible early signs of metabolic, cardiac, and neurological complications, using a whole-body MRI protocol in a population-based sample without overt cardiovascular disease. Cerebral MRI variables included gray matter (GM) volume, white matter hyperintensities (WMH), and cerebral microbleeds indicative for cerebrovascular disease [23,24] and potential neurodegeneration [25]. MRI-based fat depots included total adipose tissue (TAT), VAT, and hepatic fat as risk factors for metabolic and cardiovascular disease [26]. Among cardiac MRI parameters, stroke volume, end-systolic volume, ejection fraction, and peak ejection rate are markers of systolic cardiac function (ventricular contraction during systole). Early and late diastolic filling rates are markers for diastolic cardiac function (ventricular compliance during diastolic filling of the ventricle). End-diastolic volume and myocardial mass serve as MRI markers for myocardial remodeling (dilatation or hypertrophy of the left ventricle). Improving our understanding of the health effects of coffee drinking may have important public health implications due to the widespread exposure to coffee and the high prevalence of coronary heart disease, obesity, and neurodegenerative disease.

## 2. Materials and Methods

### 2.1. Study Design and Population

Data were derived from the population-based Cooperative Health Research in the Region Augsburg (KORA) FF4 study (2013–2014, 2279 participants), which is the second follow up of individuals who had participated in the baseline study (KORA S4, 1999–2001, 4261 participants). The KORA FF4 study included a whole-body MRI substudy (n = 400, age 39–73 years old). Details of the study protocol, MRI examination, inclusion and exclusion criteria have previously been described elsewhere [27]. Briefly, participants with prediabetes (n = 103), diabetes (n = 54), and controls (n = 243), who had no prior history of cardiovascular disease (no percutaneous coronary intervention, myocardial infarction, bypass graft, peripheral artery disease, or stroke) underwent whole-body MRI. The study was approved by the local institutional review board of the Medical faculty of the Ludwig Maximilians University, Munich, Germany.

### 2.2. Dietary Assessment

The dietary assessment comprised of repeated 24 h food lists (24HFLs) and a food frequency questionnaire (FFQ). Overall, 24HFLs included >300 food items and were used to assess food consumption over the past 24 h [28]. The FFQ, which was based on the German multilingual European Food Propensity Questionnaire, included 148 food items and was used to determine the frequency and amount of consumption over the past year [29]. Habitual food intake was computed using a blended two-step approach, based on standards of the National Cancer Institute and the Multiple Source Method, which take into account the consumption probability of a certain food on a given day and the portion size in which the food is usually consumed; the methods were described in detail elsewhere [30]. In brief, the consumption probability of the different food items included in 24HFLs and FFQ was estimated for each participant using logistic mixed models. Usual portion sizes were predicted based on data from exact 24 h dietary recalls completed by participants of the Bavarian Food Consumption Survey II, using linear mixed models and taking into account individual characteristics. Finally, habitual food intake was computed by multiplying the estimated portion size by the estimated consumption probability for each food item and participant. Habitual food intake was then categorized into different food groups and subgroups according to the classification system of the European Prospective Investigation into Cancer and Nutrition Software (International Agency for Research on Cancer, Lyon, France). The subgroup “coffee” includes all types of coffee (in g/d). Additionally, the participants’ nutrient intake was estimated by linking the habitual food intake data to the National Nutrient Database (Bundeslebensmittelschlüssel 3.02), allowing also for the calculation of total energy intake (kcal/d) and alcohol intake (in g/d).

### 2.3. Assessment of Population Characteristics

Health risk factors of the study population were assessed within the KORA study and have been previously described in detail [31]. Diabetes was defined as fasting glucose ≥7.0 mmol/l (126 mg/dl) and/or 2 h serum glucose ≥11.1 mmol/l (200 mg/dl), according to WHO recommendations [32]. Individuals were classified as smokers when they reported current, regular, or sporadic cigarette smoking and were classified as never, ex, or current smokers. Hypertension was defined as systolic blood pressure of at least 140 mmHg, diastolic blood pressure of at least 90 mmHg, or current antihypertensive treatment [33]. Other covariates included energy intake (kcal/day) and physical activity (active in summer and winter and active for ≥ 1 h per week in at least one season, inactive (= reference)) [34].

### 2.4. MR Image Acquisition

All individuals were scanned on the same three Tesla MRI scanner (Magnetom Skyra, Siemens Healthineers, Erlangen, Germany) with a whole-body radiofrequency coil–matrix system. MRI examinations were performed of the brain, cardiovascular system, and adipose tissue. 

MRI-sequences of the brain imaging protocol included 3D fluid attenuation inversion recovery (FLAIR) (TI 1800 ms, TR 5000 ms, TE 389 ms, flip angle 120°, in-plane resolution 0.5 mm, slice thickness 0.9 mm, matrix size 256 × 256, and field of view [FOV] 245 × 245 mm), and axial susceptibility-weighted imaging (SWI) (TR 27 ms, TE 20 ms, flip angle 15°, in-plane resolution 0.9 mm, slice thickness 2.5 mm, matrix size 256 × 223, and FOV 208 × 230 mm). For assessment of total and visceral adipose tissue, a two-point Dixon gradient-echo sequence was performed (TR 4.06 ms, TE 1.26, 2.49 ms, flip angle 9°, in-plane resolution 1.7 mm, slice thickness 1.7 mm, matrix size 256 × 256, and FOV 488 × 716 mm). Hepatic proton density fat fraction (PDFF_hepatic_) was quantified by using a multi-echo Dixon based on a volume interpolated body examination (VIBE) sequence (TR 8.90 ms, six TEs ranging from 1.23 to 7.38 ms, flip angle 4°, slice thickness 4 mm, matrix size 256 × 256). The cardiac imaging protocol included a four-chamber view and short-axis stack with 10 slices and 25 phases using steady-state free precession (SSFP) pulse sequences (TR 29.97 ms, TE 1.46 ms, flip angle 62° [SAX] or 63° [LAX], in-plane spatial resolution 1.5 × 1.5 mm, slice thickness 8 mm, matrix 297 × 360) [27].

### 2.5. MR Image Analysis

All image analyses were performed on offline workstations by independent readers blinded with respect to clinical status.

### 2.6. MRI-Based Intracranial Variables

A warp-based, automated brain segmentation tool was applied to 3D FLAIR images of the brain. Total gray (GM) and white matter (WM) volumes were calculated using an Automatic Anatomical Labeling Atlas [35]. In an evaluation study, the results of FLAIR-based segmentation were compared to corresponding results of warp-based brain segmentation of T1-weighted images with overall good results [36]. GM and WM brain volumes were normalized by using the ratio method Volume_corrected_ = Volume/total intracranial volume (ICV) [37]. ICV was calculated by adding up GM, WM, and cerebrospinal fluid volumes [38].

White matter hyperintensities including presence (yes/no) and age-related white matter changes scale (ARWMC) scores were graded on 3D FLAIR images. The ordinal ARWMC score ranges between 0 and 3 in each of 10 areas in total. The presence of WMH is defined as an ARWMC score >0 in any area. Total ARWMC score is the sum of scores of all areas and was treated as a continuous outcome after square-root transformation [27,39]. To determine the total volume of WMH, manual segmentation of all WMH was performed on axial FLAIR images (manuscript submitted). 

Cerebral microhemorrhages were defined as intracerebral, small areas of signal loss (≥2 mm) on axial SWI images. Symmetric signal loss in the globus pallidus, which most likely corresponds to calcifications, flow void artifacts of the cerebral blood vessels, and other intracerebral lesions with a hemorrhagic component were excluded [27].

### 2.7. MRI-Based Fat Depots

Based on the 3D VIBE-Dixon sequence, a fat selective tomogram was calculated (slice thickness 5 mm at 5 mm increment). Subcutaneous adipose tissue (SAT) from the femoral head to the cardiac apex and VAT from the level of the femoral head to the diaphragm were quantified semi-automatically using an in-house algorithm based on Matlab R2013a [40]. All segmentations were adjusted manually if necessary [27]. TAT was defined as the sum of SAT and VAT (Figure 1). Mean PDFF_hepatic_ was calculated by manually drawing a region of interest on one slice at the level of the portal vein avoiding large vessels and surrounding extrahepatic tissue [41].

### 2.8. MRI-Based Cardiac Function

Evaluation of cine-SSFP sequences on cardiac MRI to obtain left ventricular (LV) function was performed semi-automatically using commercially available software (cvi42, Circle Cardiovascular Imaging, Calgary, Alberta, Canada) [27]. The reading was performed by two alternative readers, who were blinded to any information regarding the subjects’ characteristics and according to standardized postprocessing guidelines of the Society for Cardiovascular Magnetic Resonance [42]. Automatic contour detection was performed of the LV endocardial and epicardial border and corrected manually, if necessary. The papillary muscles were excluded from the myocardial mass and included in the ventricular volumes. The end-diastolic and end-systolic phases were identified automatically by the software. LV volumetric data included end-diastolic and end-systolic volumes with calculated stroke volume (end-diastolic volume minus end-systolic volume), ejection fraction ((stroke volume/end-diastolic volume)*100), and myocardial mass. A representative example of the LV contouring is given in Figure 2. Filling and ejection rates were quantified by using dedicated in-house software (pyHeart). This software displays the LV volume versus time curve along with its derivative and estimates peak gradients during early and late LV filling due to atrial contraction [43].

### 2.9. Statistical Analysis

Demographics and clinical characteristics, such as coffee intake, were stratified by sex and summarized by mean and standard deviation for continuous variables or number and percentage for categorical variables.

Associations of coffee consumption and MRI-based markers including GM and WM brain volume, WMH volume, total, visceral and hepatic fat, and cardiac MRI parameters were evaluated using linear regression models providing β-coefficients with 95% confidence intervals for a standard deviation increase of coffee consumption (SD = 131.7 g/day). Associations of coffee intake and myocardial mass, presence of WMH or cerebral microhemorrhages on MRI were assessed by logistic regression providing odds ratios (ORs) with 95% confidence interval. Association of ARWMC score and coffee consumption was evaluated using negative binomial regression providing an incident rate ratio (IRR) with 95% confidence intervals.

Models were adjusted stepwise for age, sex (Model A), smoking status, hypertension, diabetes, Low-density lipoprotein (LDL), triglycerides (Model B), and alcohol consumption (Model C). A *p*-value of <0.05 was considered statistically significant. Assumption of linearity and normal distributions of residuals were checked visually. Multivariable spline models did not reveal nonlinear associations. Tests for effect modifications of the associations by confounding variables (age, sex, smoking status, hypertension, diabetes, LDL, triglycerides, and alcohol consumption) revealed no significant results. Statistical analyses were performed using Stata 14.1 (Stata Corporation, College Station, TX, USA).

## 3. Results

### 3.1. Population Characteristics

Among 400 enrolled subjects of the KORA FF4 cohort who underwent whole-body MRI, a total of 300 subjects had complete image acquisition, sufficient image quality of cardiac function, and complete data about coffee consumption. Demographic and risk profiles of these 300 study participants stratified by sex (132 females and 168 males with a mean age of 56.3 ± 8.8 years old or 56.2 ± 9.3 years old, respectively) are provided in Table 1. Women had significantly lower triglyceride levels and lower values of systolic/diastolic blood pressure and drank less alcohol (*p* < 0.001), compared to men. There was no significant difference in coffee consumption between men and women, with an average of approximately 393 g/day for both sexes (*p* = 0.994). For the distribution of coffee consumption, see Appendix A.

### 3.2. Associations between Coffee Consumption and Intracranial MRI Findings

Multivariable linear regression analyses showed no statistical association between coffee consumption and total brain volume in all three models. There was also no significant association between coffee drinking and WMH with regard to the presence of WMH, ARWMC score, or the total volume of detected WMH. Coffee intake was not significantly associated with cerebral microhemorrhages (Table 2).

### 3.3. Associations between Coffee Consumption and MRI-Based Fat Depots

Table 3 shows the association between coffee drinking and different fat depots measured on MRI. VAT was inversely associated with coffee consumption when adjusting for age and sex (β = −0.32 (95%CI: −0.57; −0.06), *p* < 0.05) and after additional adjustment for smoking, hypertension, diabetes, LDL and triglycerides (β = −0.23 (95%CI: −0.45; −0.01), *p* < 0.05). However, VAT was no longer statistically associated with coffee intake after additional adjustment for alcohol consumption (β = −0.20 (95%CI: (−0.43; 0.02), *p* > 0.05). In all three models, there was no significant association between coffee consumption and TAT or PDFF_hepatic_ (Table 3).

### 3.4. Associations between Coffee Intake and Cardiac MRI Parameters

Coffee intake was positively associated with MR-based cardiac function parameters, such as late diastolic filling rate (β = 22.69 [95%CI: 7.73; 37.65], *p* < 0.01), stroke volume (β = 1.56 [95%CI: 0.48; 2.64], *p* < 0.01) and ejection fraction (β = 0.94 [95%CI: 0.08; 1.80], *p* < 0.05) when adjusting for age and sex. All three parameters remained significantly associated with coffee consumption after additional adjustment for smoking, hypertension, diabetes, LDL, triglycerides, and also after additional adjustment for alcohol consumption. There was no significant association between coffee consumption and early diastolic filling rate, end-diastolic volume, end-systolic volume and myocardial mass (Table 4). 

### 3.5. Additional Analysis of VAT and Cardiac Function

Further adjustment of the multivariable linear regression analyses between coffee intake and VAT included overall energy consumption (kcal/day) and physical activity but did not change the result substantially with β = −0.18 (−0.40; 0.04) *p* = 0.107, compared to Model C (see Table 3). Recalculating the simple Model A with adjustment for only age, sex, overall energy intake, and physical activity, revealed the following significant result: β = −0.26 (−0.50; −0.01), *p* = 0.041. Adding BMI to Model C did not substantially change the association between coffee consumption and cardiac function. Additional analyses with tertiles of coffee consumption comparing low, middle, and high coffee consumption were performed for cardiac function. These analyses revealed a significantly higher ejection fraction for the high coffee intake tertile, compared to the low coffee intake tertile (β = 2.22 (95%CI: 0.01; 4.43), *p* < 0.05). No other cardiac parameters differed among the three coffee intake groups (see Appendix A). 

## 4. Discussion

In this population-based cohort study, subclinical health effects of coffee drinking were investigated using MRI-based markers of potential neurodegeneration, cardiac function, and fat depots. Our data indicate a positive relationship between coffee intake and systolic and diastolic cardiac function, independent of age, sex, smoking, hypertension, diabetes, LDL, triglycerides, and alcohol consumption. Coffee consumption was inversely associated with VAT independent of demographic variables and cardiovascular risk factors, but this association did not remain significant after additional adjustment for alcohol consumption. We found no significant association between coffee drinking and MRI markers for potential neurodegeneration.

The exact mechanisms of how coffee consumption affects our health are still not fully understood, which may be due to the complex mixture of different bioactive compounds within roasted coffee [44,45]. On the one hand, the protective effects of coffee intake include the reduction of oxidative stress and inflammatory markers [46]. On the other hand, coffee consumption seems to increase serum triglycerides, cholesterol, and low-density lipoprotein [15]. It is possible that it is due to these opposing health effects that we did not find any significant evidence for a beneficial impact of coffee regarding potential neurodegeneration. These findings are partly in accordance with a study embedded within the population-based Rotterdam Study, which also did not observe a significant association between coffee consumption and MRI-based markers for potential neurodegeneration, including GM and WM brain volume and WMH [47]. However, the authors describe that higher coffee consumption was associated with a lower prevalence of lacunar infarcts [47], which we did not evaluate in our study. The only study investigating the relationship between intracerebral microhemorrhages and coffee intake suggests that habitual coffee consumption reduces microhemorrhage risk in men but not in women [48]. This sex difference was independent of other demographic and lifestyle variables, including smoking, alcohol consumption and BMI. Our data did not suggest a significant association between coffee drinking and intracerebral microhemorrhages, independent of sex among other variables. 

Adiponectin and leptin are both important adipokines in the regulation of energy balance [49]. Decreased levels of adiponectin [50] and increased levels of leptin have both been associated with increased visceral and hepatic fat content [51,52]. In a cross-sectional analysis of Japanese men and women, coffee intake was positively associated with adiponectin and inversely associated with leptin levels [53]. This favorable link between coffee consumption and adipokines might explain why coffee drinking was inversely associated with TAT, VAT, and PDFF_hepatic_ in our study. This inverse association remained significant only for VAT after adjusting for age, sex, smoking, hypertension, diabetes, LDL, and triglycerides but not after additional adjustment for alcohol consumption. Interestingly, increased VAT is more closely associated with an adverse cardiometabolic profile than TAT [54], liver, or subcutaneous fat [55]. Similar to our results, a population-based study of the PopGen Biobank in Kiel, northern Germany, found a significant inverse association between coffee intake and the ratio of VAT and subcutaneous abdominal adipose tissue assessed by MRI [52]. In this study, the regression model was adjusted for sex, age, physical activity and total energy intake but not for alcohol consumption. Therefore, it cannot be ruled out that reduced VAT among coffee drinkers arises from combined residual confounding since coffee consumption correlates with other lifestyle markers such as alcohol [56,57].

Several studies found an association between coffee consumption and a lower risk of cardiovascular disease [4]. Potential protective effects from coffee drinking are even reported following acute myocardial infarction with a reduced risk of mortality [44] and, at low doses, in patients with atrial fibrillation [58]. The positive long-term effects of coffee on cardiovascular disease include improvement of endothelial function, increased insulin sensitivity, and stimulation of fatty acid oxidation in the liver [59]. There are several mechanisms that might underly the effect of coffee consumption on cardiac health. Coffee contains several bioactive substances, which can modulate antioxidant effects [59]. Oxidant stress has been linked to the pathogenesis of atherosclerosis and the incident of coronary artery disease [60]. Coffee is also a source of a bioactive compound called flavonoid. Experimental studies suggest that flavonoids modulate gene expression and signaling pathways, for example, by activating 5-monophosphate-activated protein kinase, by increasing Toll-like receptor-4 expression, increasing nitrite oxidase synthase activity, and reducing the production of reactive oxygen species [61]. These molecular mechanisms induce other effects, including decreased arterial stiffness, lower insulin resistance, and improved glycemic control, resulting in a lower risk of metabolic syndrome and diabetes mellitus [62]. Furthermore, caffeine has a mild and transitory vasoconstrictor effect; however, its main and predominant effect is vasodilatation. Caffeine acts on vascular smooth muscle cells directly and indirectly, and on endothelial cells [63]. Despite the vast body of literature regarding the effect of coffee consumption on cardiovascular health, there are few data on the direct effects of coffee consumption on cardiac function itself. One study found no acute effect on cardiac function after oral caffeine intake, assessed by echocardiography [64]. Two studies using impedance cardiography found that oral caffeine exposure resulted in an acute increase of blood pressure and vascular resistance but no relevant increase in cardiac output [65,66]. This discrepancy to the results of our study might be explained by the differences in the technique used and sample size, and exposure variable since there is evidence that coffee might be superior to caffeine alone in providing beneficial effects [67,68].

Several other studies reported on the effects of coffee intake on cardiac MRI measures yet with the focus on perfusion measurements on stress cardiac MRI [69,70,71]. Interestingly, Greulich et al. showed that two cups of coffee one hour before the examination caused a significant decrease in the ischemic stress, compared to caffeine-naïve adenosine stress cardiac MRI, probably due to coronary hyperemia [72]. The one previous study that analyzed the impact of coffee on cardiac function based on cardiac MRI in 10 subjects found a significant decrease in left ventricular end-diastolic volume but no significant change of left ventricular stroke volume and ejection fraction one hour after coffee consumption, compared to baseline [73]. The apparently differing results of this preliminary study with a small sample size compared to our study might be explained, inter alia, by the fact that they focused on acute effects of coffee consumption, whereas it is most likely that our study looked at long-term effects.

Among the strengths of our study are the population-based study design with a large sample size and the focus on MRI-based markers of potential neurodegeneration, cardiac function, and fat depots, with cardiac MR currently being considered the reference standard for assessment of global myocardial function. However, there are some limitations of our study that need to be taken into account when interpreting the data. First, we could not account for the coffee preparation method (e.g., filtered or unfiltered coffee, strength of the brew, and sugar content) or the type of coffee. Yet, it should be noted that coffee is a very complex mixture with several hundreds of chemical compounds, highlighting the impossibility to study the effect of each of them separately. Secondly, we did not investigate the association of coffee drinking and late gadolinium enhancement (LGE), because the number of subjects with positive LGE on cardiac MRI in our study cohort (n = 9) was too small for statistical analysis. This might also apply to the effect of heavy coffee consumption; a recent meta-analysis, which summarized evidence from 36 prospective studies, involving a total of 1,279,804 participants, suggests a U-shaped association between habitual coffee intake and CVD risk [74]. Our study could not corroborate this finding, which is most probably due to the small number of heavy coffee drinkers in our study. Further investigations with a larger scale are needed to evaluate potential negative effects of heavy coffee intake on MRI-based markers of a whole-body MRI protocol and whether coffee consumption has an impact on LGE found on cardiac MRI. Furthermore, the results were not adjusted for multiple comparisons, and thus, weak associations remain questionable. However, an adjustment of the significance level for the seven tests of the association between coffee intake and cardiac function parameters (using the Bonferroni method) revealed a consistent association between coffee intake and stroke volume (*p* < 0.05/7). Finally, the MRI sub-study was originally designed as a nested case–control study comparing subjects with diabetes, prediabetes, and non-diabetic controls. Consequently, the study population of this MRI substudy, which included the whole cohort, is not representative due to the enrichment of prediabetic and diabetic subjects. However, additional analyses using weights accounting for differences between the MRI substudy and the whole study cohort did not reveal any substantial dissimilarity [75].

## 5. Conclusions

In conclusion, our study of a population-based cohort without overt cardiovascular disease demonstrated a positive relationship between coffee intake and MRI-based systolic and diastolic cardiac function, independent of demographic variables, cardiovascular risk factors, and several lifestyle markers.

## Figures and Tables

**Figure 1 nutrients-13-01275-f001:**
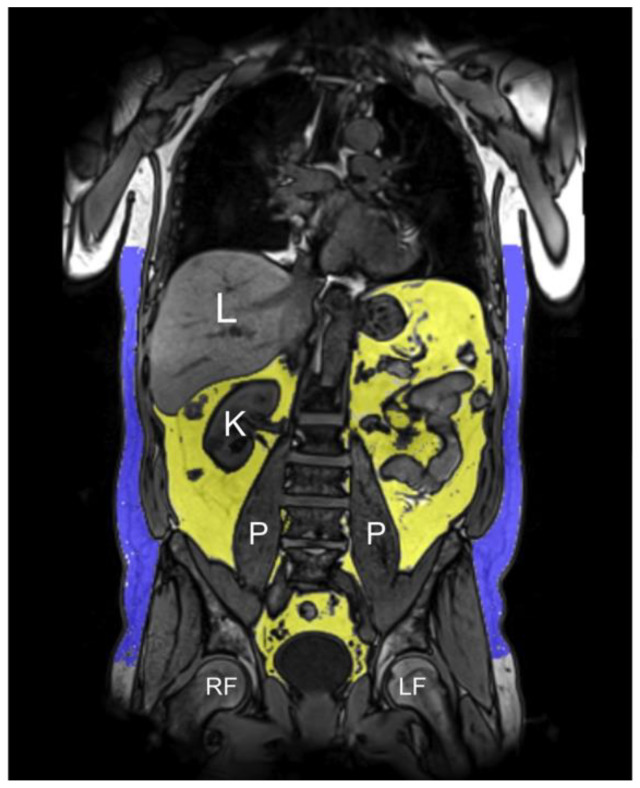
Representative example of the semi-automatic MRI-based assessment of fat depots in a 60-years-old male on reconstructed 3D VIBE-Dixon images (VAT 5.3 L, SAT 7.7 L, TAT 13.0 L). Visceral adipose tissue (VAT, yellow area) and subcutaneous adipose tissue (SAT, blue area) were measured from the diaphragm/cardiac apex to the femoral head; total adipose tissue (TAT) was defined as the sum of VAT and SAT, indicated in liter. L = liver, K = kidney, P = psoas muscle, RF = right femoral head, LF = left femoral head.

**Figure 2 nutrients-13-01275-f002:**
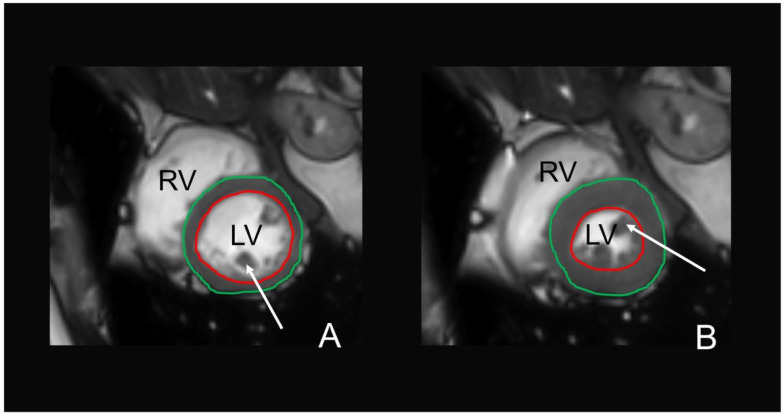
MRI-based assessment of the left ventricular function on cine steady-state free precession (cine-SSFP) sequences in a 56-years-old female. The endocardial (red contour) and epicardial (green contour) border were detected automatically in end-diastole (**A**) and end-systole (**B**) and manually corrected, if necessary. The papillary muscles (white arrows) were included in the left ventricular lumen. LV = left ventricle, RV = right ventricle.

**Table 1 nutrients-13-01275-t001:** Characteristics of the study population, stratified by sex.

	Total n = 300	Women n = 132	Men n = 168	*p*-Value *
Age (years)	56.3 ± 9.1	56.3 ± 8.8	56.2 ± 9.3	0.967
Coffee intake (g/day)	392.5 ± 131.7	392.6 ± 118.8	392.5 ± 141.4	0.994
LDL (mg/dl)	138.9 ± 33.4	135.8 ± 33.1	141.4 ± 33.6	0.152
Triglycerides (mg/dl)	126.0 ± 78.7	101.6 ± 43.5	145.1 ± 93.6	**<0.001**
Smoking status				0.103
Never-smoker	109 (36.3%)	55 (41.7%)	54 (32.1%)	
Ex-smoker	134 (44.7%)	50 (37.9%)	84 (50.0%)	
Current smoker	57 (19.0%)	27 (20.5%)	30 (17.9%)	
Alcohol consumption (g/day)	18.5 ± 24.1	7.8 ± 13.7	26.9 ± 27.0	**<0.001**
Diabetes mellitus	33 (11%)	11 (8.3%)	22 (13.1%)	0.198
Systolic BP (mmHg)	119.8 ± 16.4	112.8 ± 14.4	125.4 ± 15.8	**<0.001**
Diastolic BP (mmHg)	74.8 ± 9.9	71.7 ± 8.7	77.3 ± 10.1	**<0.001**

Values in mean ± standard deviation or number and percentage, * *p*-values are from *t*-test or chi^2^-test, *p*-values marked in bold were significant, n = 300. LDL = Low-density lipoprotein, BP = blood pressure.

**Table 2 nutrients-13-01275-t002:** Associations between coffee drinking and cerebral MRI findings.

Coffee Intake (g/day)	Model A	Model B	Model C
Per 1 SD increment	β (95%CI)	β (95%CI)	β (95%CI)
Gray matter volume	−0.0006 (−0.0022; 0.001)	−0.0006 (−0.0022; 0.001)	−0.0006 (−0.0022; 0.001)
White matter hyperintensities	−0.0012 (−0.0029; 0.0006)	−0.0012 (−0.0029; 0.0006)	−0.0011 (−0.0029; 0.0006)
WMH volume	403 (−237.4; 1043.4)	448.4 (−205.9; 1102.6)	381.4 (−275.7; 1038.6)
Presence of WMH (yes/no)	OR: 0.97 (0.75; 1.26)	OR: 0.99 (0.75; 1.30)	OR: 0.99 (0.75; 1.31)
ARWMC score	IRR: 1.05 (0.92; 1.20)	IRR: 1.07 (0.93; 1.22)	IRR: 1.06 (0.92; 1.22)
Cerebral microbleeds	OR: 1.08 (0.76; 1.55)	OR: 1.10 (0.75; 1.60)	OR: 1.10 (0.75;1.61)

β-coefficients are from linear regression models (odds ratio (OR) from logistic regression, incident rate ratio (IRR) from negative binomial regression) adjusted for age and sex (Model A), as Modal A and additionally adjusted for smoking, hypertension, diabetes, LDL, triglycerides (Model B) and as Model A and B and additionally adjusted for alcohol consumption (Model C), n = 276. WMH = White matter hyperintensities, ARWMC = white matter changes scale, IRR = incident rate ratio, OR = odds ratio.

**Table 3 nutrients-13-01275-t003:** Associations between coffee consumption and adiposity markers.

Coffee Intake (g/day)	Model A	Model B	Model C
Per 1 SD increment	β (95%CI)	β (95%CI)	β (95%CI)
TAT	−0.07 (−0.71; 0.56)	0.09 (−0.48; 0.67)	0.08 (−0.50; 0.66)
VAT	**−0.32 (−0.57; −0.06)** *	**−0.23 (−0.45; −0.01)** *	−0.20 (−0.43; 0.02)
PDFF_hepatic_	−0.62 (−1.52; 0.29)	−0.33 (−1.15; 0.48)	−0.26 (−1.09; 0.56)

β-coefficients are from linear regression models adjusted for age and sex (Model A), as Model A and additionally adjusted for smoking, hypertension, diabetes, LDL, triglycerides (Model B) and as Model A and B and additionally adjusted for alcohol consumption (Model C), n = 299–314 * *p* < 0.05. TAT = total adipose tissue, VAT = visceral adipose tissue, PDFF_hepatic_ = Hepatic proton density fat fraction.

**Table 4 nutrients-13-01275-t004:** Associations between coffee intake and cardiac MRI parameters.

Coffee Intake (g/day)	Model A	Model B	Model C
Per 1 SD increment	β (95%CI)	β (95%CI)	β (95%CI)
Early diastolic filling rate (ml/s)	7.03 (−5.27; 19.32)	5.61 (−6.51; 17.72)	4.47 (−7.73; 16.67)
Late diastolic filling rate (ml/s)	**22.69 (7.73; 37.65)** **	**18.93 (4; 33.87)** *	**18.19 (3.12; 33.27)** *
End-diastolic volume (ml/m^2^)	1.29 (−0.35; 2.94)	1.17 (−0.43; 2.76)	1.08 (−0.54; 2.69)
End-systolic volume (ml/m^2^)	−0.26 (−1.16; 0.64)	−0.32 (−1.22; 0.58)	−0.35 (−1.26; 0.56)
Stroke volume (ml/m^2^)	**1.56 (0.48; 2.64)** **	**1.49 (0.45; 2.53)** **	**1.44 (0.39; 2.48)** **
Ejection fraction (%)	**0.94 (0.08; 1.80)** *	**0.98 (0.10; 1.85)** *	**0.97 (0.09; 1.86)** *
Peak ejection rate (ml/s)	−9.78 (−24.52; 4.96)	−6.97 (−21.58; 7.64)	−5.33 (−20.03; 9.36)
Myocardial mass (g/m^2^)	0.44 (−0.9; 1.77)	0.84 (0.47; 1.51)	0.84 (0.47; 1.52)

β-coefficients are from linear regression models adjusted for age and sex (Model A), as Model A and additionally adjusted for smoking, hypertension, diabetes, LDL, triglycerides (Model B) and as Model A and B and additionally adjusted for alcohol consumption (Model C), n = 300, * *p* < 0.05; ** *p* < 0.01.

## Data Availability

The datasets used and analyzed in the current study are available from the corresponding author on reasonable request.

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
