# Peer review of "Significant Impact of Coffee Consumption on MR-Based Measures of Cardiac Function in a Population-Based Cohort Study without Manifest Cardiovascular Disease"

_nutrients, 2021, doi:10.3390/nu13041275_

Round 1

Reviewer 1 Report

Due to the popularity of coffee studies on the relationship between coffee drinking habit and the risk for cardiovascular and neurodegenerative diseases are necessary and very interesting  

However, the article submitted for review requires corrections and explanations by the authors, especially in the research methodology.

- it is not understandable what groups of people were compared. The authors briefly inform that there were three groups of subjects: prediabetes, diabetes and controls (implicitly people without diabetes). The results of the study show that all these people drank coffee, so how was the MRI assessed (compared) if there were no people who did not drink coffee? In order to capture the changes that would result from coffee drinking, the results of the study in coffee drinkers should be compared with the results of non-coffee drinkers. Did the study only compare the effects of drinking coffee in people with and without diabetes? The methodology of the study is not understood!

- on what basis do the authors believe that coffee drinking was inversely associated with visceral adipose tissue, if they did not take into account the energy value of the subjects' diet or their physical activity?

- it is nowhere given for how many years the participants regularly drank coffee and whether there were heavy coffee drinkers and little coffee drinkers in this group

- the downside of the study is the lack of analysis of the type of coffee and coffee preparation method (including filtered or unfiltered coffee, strength of the brew, added sugar)

- in the Introduction (line 51), very old studies (1999) are cited on the negative effects of coffee on blood pressure, cholesterol and homocysteine ​​levels. Perhaps more recent studies have denied such a link. Authors should be more objective and include the latest research here as well

Technical remarks:

- the amount of coffee consumption should be given in "ml/day" rather than in "g/day" (e.g. 393 g/day, etc.)

- in line 251 instead of Table 2 there should be Table 4

Author Response

Please see the attachment: 

Due to the popularity of coffee studies on the relationship between coffee drinking habit and the risk for cardiovascular and neurodegenerative diseases are necessary and very interesting  

Author Response: We thank the reviewer for the favorable evaluation of our manuscript.

However, the article submitted for review requires corrections and explanations by the authors, especially in the research methodology.

R1.1: it is not understandable what groups of people were compared. The authors briefly inform that there were three groups of subjects: prediabetes, diabetes and controls (implicitly people without diabetes). The results of the study show that all these people drank coffee, so how was the MRI assessed (compared) if there were no people who did not drink coffee? In order to capture the changes that would result from coffee drinking, the results of the study in coffee drinkers should be compared with the results of non-coffee drinkers. Did the study only compare the effects of drinking coffee in people with and without diabetes? The methodology of the study is not understood!

Author Response R 1.1: Thank you very much for pointing out this lack of clarity.

Action R 1.1: For the main analyses, we decided to investigate coffee consumption as a continuous variable and to estimate the effect for each standard deviation (g/day) increase for the whole cohort (prediabetes, diabetes and controls). However, we understand the interest in comparing different coffee drinker groups. Therefore, the following information and Supplementary Table1 have been added to the results section (Page 8):

“Additional analyses with tertiles of coffee consumption comparing low, middle and high coffee-consumption were performed for cardiac function. These analyses revealed a significantly higher ejection fraction for the high coffee intake tertile compared to the low coffee intake tertile (β=2.22 [95%CI: 0.01; 4.43], p<0.05). No other cardiac parameters differed among the three coffee intake groups (see Table S1).

Table S1: MRI based cardiac function comparing low, middle and high coffee-consumption

Coffee intake

(g/day)

Middle Tertile

(383.6-458.4)

High Tertile

(459.0-699.2)

β (95%CI)

β (95%CI)

Early diastolic filling rate (ml/s)

-4.99 (-34.6; 24.62)

1.20 (-29.21; 31.61)

Late diastolic filling rate (ml/s)

33.43 (-3.27; 70.12)

28.58 (-9.11; 66.26)

End-diastolic volume  (ml/m2)

-0.26 (-4.20; 3.68)

0.88 (-3.16; 4.93)

End-systolic volume (ml/m2)

-1.34 (-3.55; 0.88)

-1.26 (-3.53; 1.02)

Stroke volume (ml/m2)

1.02 (-1.55; 3.59)

2.14 (-0.50; 4.79)

Ejection fraction (%)

1.89 (-0.26; 4.04)

2.22 (0.01; 4.43)*

Peak ejection rate (ml/s)

-9.55 (-45.19; 26.09)

2.92 (-33.68; 39.53)

Myocardial mass (g/m2)

2.57 (-0.69; 5.82)

1.91 (-1.43; 5.25)

Reference group: low tertile of coffee intake (39.5-383.2 g/day)). β-coefficients are from linear regression models adjusted for age and sex, smoking, hypertension, diabetes, LDL, triglycerides and alcohol consumption, n=300, *p<0.05; **p<0.01.

Please see also following changes in the limitation section (Page 10):

“Finally, the MRI sub-study was originally designed as a nested case-control study comparing subjects with diabetes, prediabetes and non-diabetic controls. Consequently, the study population of this MRI sub-study, which included the whole cohort, is not representative due to enrichment of prediabetic and diabetic subjects. However, additional analyses using weights accounting for differences between the MRI sub-study and the whole study cohort did not reveal any substantial dissimilarity (75).”

R1.2: On what basis do the authors believe that coffee drinking was inversely associated with visceral adipose tissue, if they did not take into account the energy value of the subjects' diet or their physical activity?

Author Response R 1.2: We thank the reviewer for the opportunity to clarify this result. We investigated the association between coffee intake and VAT in several models with a stepwise increase of possible confounders that were taken into account. In the first two models with limited confounders, we detected an inverse relation between coffee intake and VAT. However, in the fully adjusted model (including alcohol consumption) we could not detect a significant association between coffee intake and VAT.

Action R 1.2: We have adapted the results section and table 3 of the manuscript accordingly (page 8):

“Further adjustment of the multivariable linear regression analyses between coffee intake and VAT included overall energy consumption (kcal/day) and physical activity, but did not change the result substantially with β=-0.18 [-0.40; 0.04] p=0.107 compared to Model C (see table 3). Recalculating the simple model with adjustment for only age, sex, overall energy intake and physical activity, revealed the following significant result: β=-0.26 (-0.50; -0.01) p=0.041.“

We have added the following information to the “Assessment of Population Characteristics” segment in the materials and methods section (page 3):

“Other covariates included energy intake (kcal/day) and physical activity [active in summer and in winter and active for ≥ 1 h per week in at least one season, inactive (= reference)] (37).”

R1.3: It is nowhere given for how many years the participants regularly drank coffee and whether there were heavy coffee drinkers and little coffee drinkers in this group.

Author Response R 1.3:  We appreciate this comment. Coffee consumption was measured and calculated for the last 12 months. Information on lifelong coffee consumption was not available.

Action R 1.3: We have provided the distribution of coffee intake that displays the overall range and the ranges and proportions for heavy and little coffee drinkers (Supplementary Figure 1). Furthermore, the following information has been added to the results section (page 6):

“For the distribution of coffee consumption please see Figure S1.”

R1.4: The downside of the study is the lack of analysis of the type of coffee and coffee preparation method (including filtered or unfiltered coffee, strength of the brew, added sugar).

Author Response R 1.4: We agree with the reviewer that it would be interesting to include more detailed information on the preparation method. However, this information is not available for the KORA cohort.

Action R 1.4: We have have pointed out this limitation in the limitation section of the discussion (page 10):

“First, we could not account for the coffee preparation method (e.g. filtered or unfiltered coffee, strength of the brew and sugar content) or the type of coffee.”

R1.5: in the Introduction (line 51), very old studies (1999) are cited on the negative effects of coffee on blood pressure, cholesterol and homocysteine ​​levels. Perhaps more recent studies have denied such a link. Authors should be more objective and include the latest research here as well.

Author Response R 1.5: Thank you very much for this suggestion.

Action R 1.5: The sentence has been rephrased (page 2) and the following references have been included:

“However, there are conflicting results whether intake of coffee is associated with adverse effects on blood pressure (11-13). Nevertheless, most studies suggest that coffee consumption is associated with adverse effects on blood cholesterol (16,17) and homocysteine levels (18).”

References:

  1. Palatini P, Dorigatti F, Santonastaso M, Cozzio S, Biasion T, Garavelli G, Pessina AC, Mos L. Association between coffee consumption and risk of hypertension. Ann Med 2007;39:545-53.
  2. Noordzij M, Uiterwaal CS, Arends LR, Kok FJ, Grobbee DE, Geleijnse JM. Blood pressure response to chronic intake of coffee and caffeine: a meta-analysis of randomized controlled trials. J Hypertens 2005;23:921-8.
  3. Steffen M, Kuhle C, Hensrud D, Erwin PJ, Murad MH. The effect of coffee consumption on blood pressure and the development of hypertension: a systematic review and meta-analysis. J Hypertens 2012;30:2245-54.

...

  1. Strandhagen E, Thelle DS. Filtered coffee raises serum cholesterol: results from a controlled study. Eur J Clin Nutr 2003;57:1164-8.
  2. Cai L, Ma D, Zhang Y, Liu Z, Wang P. The effect of coffee consumption on serum lipids: a meta-analysis of randomized controlled trials. Eur J Clin Nutr 2012;66:872-7.
  3. Verhoef P, Pasman WJ, Van Vliet T, Urgert R, Katan MB. Contribution of caffeine to the homocysteine-raising effect of coffee: a randomized controlled trial in humans. Am J Clin Nutr 2002;76:1244-8.

Technical remarks:

R1.6: the amount of coffee consumption should be given in "ml/day" rather than in "g/day" (e.g. 393 g/day, etc.)

Author Response R 1.6: We appreciate this comment. However, since we did not differentiate between different types of coffee, e.g. Cold Brew and Espresso Coffees, the concentration of coffee per ml might vary significantly. E.g. a standard cup (30 ml) of Espresso coffee contains about 100 mg of caffeine, while a 225 ml cup of boiled or filtered coffee contains about 135 mg of caffeine. Therefore we concluded that the amount of coffee consumption using g/day is more representative than using ml/day. Please see also Action R 1.4.

R1.7: in line 251 instead of Table 2 there should be Table 4

Author Response R 1.7: Thank you very much for pointing this out.

Action R 1.7: Table 2 has been changed to Table 4 (page 7).

Reviewer 2 Report

Thank you for the nice manuscript. I have some suggestion that I believe would be very informative:

  • According to the methods section, no correction for multiple testing was attempted. There is also no mention of this in the limitations section. Could you comment on this and indicate whether any of the reported associations would survive multiple testing correction for example FDR.
  • The results of the MRI based cardiac function parameters are highly interesting. They point towards coffee having a potential stimulatory effect on atrial and ventricular muscles. Did you consider additionally adjusting the models examining the effect of coffee consumption and cardiac function parameters for BMI to examine and exclude residual confounding?

Reviewer 3 Report

This study deals with the interesting data. This paper is well-written on an MRI exam. There is something I’d like to ask.

What dose each index, such as GM and WMH volume, filling rate, stroke volume, refer to? Which signs are these indexes associated with?

What mechanism coffee effects on cardiac function?

Round 2

Reviewer 1 Report

I accept the authors' explanations and the article proofreading. The introduced changes make the article much more understandable.

I have no further objections